# A genome-scale map of DNA methylation turnover identifies site-specific dependencies of DNMT and TET activity

Paul Adrian Ginno [1,8], Dimos Gaidatzis[1,2,8], Angelika Feldmann[1,5,8], Leslie Hoerner[1], Dilek Imanci[1,6], Lukas Burger [1,2], Frederic Zilbermann[1], Antoine H. F. M. Peters[1,3], Frank Edenhofer[4], Sébastien A. Smallwood[1], Arnaud R. Krebs[1,7] & Dirk Schübeler [1,3✉]

DNA methylation is considered a stable epigenetic mark, yet methylation patterns can vary during differentiation and in diseases such as cancer. Local levels of DNA methylation result from opposing enzymatic activities, the rates of which remain largely unknown. Here we developed a theoretical and experimental framework enabling us to infer methylation and demethylation rates at 860,404 CpGs in mouse embryonic stem cells. We find that enzymatic rates can vary as much as two orders of magnitude between CpGs with identical steady-state DNA methylation. Unexpectedly, de novo and maintenance methylation activity is reduced at transcription factor binding sites, while methylation turnover is elevated in transcribed gene bodies. Furthermore, we show that TET activity contributes substantially more than passive demethylation to establishing low methylation levels at distal enhancers. Taken together, our work unveils a genome-scale map of methylation kinetics, revealing highly variable and context-specific activity for the DNA methylation machinery.

[1] Friedrich Miescher Institute for Biomedical Research, Basel, Switzerland. [2] Swiss Institute of Bioinformatics, Basel, Switzerland. [3] Faculty of Sciences, University of Basel, Basel, Switzerland. [4] Leopold-Franzens-University Innsbruck & CMBI, Innsbruck, Austria. [5] Present address: Department of Biochemistry, University of Oxford, Oxford, UK. [6] Present address: Novartis Institutes for Biomedical Research, Basel, Switzerland. [7] Present address: EMBL Heidelberg, Heidelberg, Germany. [8] These authors contributed equally: Paul Adrian Ginno, Dimos Gaidatzis, Angelika Feldmann. ✉email: dirk@fmi.ch

DNA methylation is a well-studied epigenetic mark in mammals, where it plays critical roles in the context of genomic imprinting, chromatin architecture, and gene regulation[1–4]. While methylation maps in mouse and human cells have provided valuable information regarding the genomic distribution of this mark[5,6], they do not reveal the actual dynamics of this DNA modification. This creates a gap in our understanding of not only how methylation patterns are actually achieved, but on their stability, a property required for them to impart long-term epigenetic effects. Indeed, actual activity can only be measured upon acute disruption of one pathway, coupled with time-resolved measurements at high resolution and the appropriate analytical framework.

The methylation machinery can be divided into the de novo methylation enzymes DNMT3a and DNMT3b[7,8] and the maintenance methyltransferase DNMT1[9,10]. All members of the Dnmt family are essential for mammalian development[8,11]. Conversely, DNA methylation can be lost either by incomplete maintenance following replication, referred to as passive demethylation[12], or actively via the ten–eleven translocation (TET) family of dioxygenase enzymes[13–17]. The TET family consists of three proteins in mice, TET1/2/3, with TET2 likely responsible for the majority of hydroxymethylation in embryonic stem cells (ESCs)[18]. Active demethylation by the TET family is thought to occur through successive oxidation of the methyl group on CpG dinucleotides[19–21], culminating in excision through the base excision repair pathway[13]. While absence of distinct members of the TET family is permissible for pluripotency and embryogenesis[22,23], reduction in TET activity has been shown to impact differentiation[24–26]. Importantly, loss of all TET enzymes is incompatible with embryogenesis[27], indicating a critical role for these proteins in differentiation and lineage specification.

While the division of labor between de novo and maintenance methylation predominantly describes DNMT3 and DNMT1 activities, respectively, evidence exists suggesting this distinction is not absolute. For example, loss of DNMT3a and DNMT3b leads to progressive loss of DNA methylation over many cell passages[28]. Recent work has also demonstrated that DNMT1 can display de novo activity in oocytes upon UHRF1 mislocalization and loss of Stella[29]. Regardless, the association of DNMT1 with the replication fork[30], the loss of 90% methylation in its absence[31], the autoinhibitory function of its CXXC domain[32], and much higher preference for hemimethylated substrates[33,34] all clearly suggest its predominant function in somatic cells is maintenance.

Previous work has sought to determine methylation activities empirically at CpG sites in vitro[35] and in cultured cells[34,36], as well as theoretically[37–39]. These studies have revealed several properties of the enzymes responsible for depositing these marks, from presence of non-CpG methylation[34,35] to the inference of methylation and maintenance rates for individual CpGs[37], as well as DNMT1 processivity[38]. More recently, these models have been extended and adapted with the aim of describing population methylation dynamics[40–42]. While informative in their own right, their genomic scope is limited or they do not quantitatively infer the rates of all three processes at the individual CpG level, including de novo and maintenance methylation, as well as active demethylation.

Here, we combine acute and stable genetic ablations of methylating and demethylating enzymes with high-coverage quantitative measurements of dynamic DNA methylation over time. Dynamical modeling of the resulting datasets enables us to infer actual rates of methylation and demethylation for individual CpGs at the scale of the genome. Our work not only profiles kinetics of methylation, but also reveals that methylation and demethylation rates are highly context specific, implicating disparate chromatin processes in shaping methylome dynamics in ESCs.

## Results

### A dynamical model and cellular system to infer turnover rates.

DNA methylation is a dynamic process, resulting in the average methylation patterns observed in various cell types. Building on previous conceptual work[41,43], these methylation averages result from opposing activities of enzymes that apply and remove DNA methylation (Fig. 1a). Here, we set out to quantify these two activities at the CpG level, namely the rate of methylation ($k_{me}$) and the rate of demethylation ($k_{de}$). We define $k_{me}$ as the rate, whereby an unmethylated cytosine (C) is converted to a methylated cytosine (5mC), while $k_{de}$ is the rate at which 5mC is converted to C. Eventually steady state is reached where the number of conversion events in both directions per unit time is equal. These equilibrium methylation levels will henceforth be referred to merely as 'steady state' (Fig. 1b). For example, if 50% methylation is measured at steady state, this means that $k_{me}$ and $k_{de}$ are equal. Viewing DNA methylation through this lens provides an explanation for population methylation levels. For example, a 75% methylated cytosine is subject to higher $k_{me}$ than $k_{de}$, while a 25% methylated cytosine is exactly the opposite (Fig. 1b). However, different rate values can give rise to the same average methylation level, as long as the ratio of the two rates remains unchanged (as shown for 50% methylated cytosines in Fig. 1b). This implies that such methylation dynamics would be masked by simply measuring average methylation levels.

Enzymatically, we attribute $k_{me}$ to the combined activity of the de novo methyltransferases DNMT3a and DNMT3b. In contrast, $k_{de}$ encompasses both active and passive demethylation, governed by the TET1/2/3 proteins and imperfect maintenance by DNMT1, respectively. In presentation of the modeling here, we use $k_{me}$ and $k_{de}$ to outline the processes, while below we will refer to the activities by more general nomenclature, namely as "de novo methylation rate", "passive demethylation rate", and "active demethylation rate". It is important to stress here that "passive demethylation" in this context is synonymous with DNMT1 infidelity. Moreover, we use the term turnover to reflect different absolute rate combinations given an identical steady state. Using the examples of 50% methylated CpGs in Fig. 1b, the CpG on the left would have higher turnover than the CpG on the right. This property can only be revealed by acute pathway disruption coupled with time-resolved measurements (Fig. 1c).

To discriminate between active and passive demethylation processes, we genetically removed all three TET enzymes, leaving DNMT1 fidelity as the sole factor influencing $k_{de}$ (Fig. 1a). Using an existing CRISPR design[44], we mutated all six Tet alleles in mouse ESCs, causing frameshifts of the proteins to create catalytically dead enzymes (Supplementary Fig. 1a). This Tet Triple Knockout (TTKO) ESC line proliferates normally as previously shown[27] and while it retains 5mC signal, hydroxymethylation is lost as observed by slot blotting with a sensitive 5hmC antibody (Supplementary Fig. 1b).

The knockout of all Tet genes was performed in a particular genetic background that enables inducible removal of de novo methylation. More specifically, we adapted an existing conditional knockout system by breeding mice where catalytic exons for both alleles of Dnmt3a and Dnmt3b are functional, but flanked by loxP sites[45,46] (Supplementary Fig. 1c, d). From these mice, we generated a stem cell line homozygous for both alleles to allow genetic deletion by the Cre recombinase (Fig. 1d). While we initially attempted to excise the fragments via inducible Cre activity, this was hindered by premature deletion events due to leaky recombinase activity (data not shown). To circumvent this

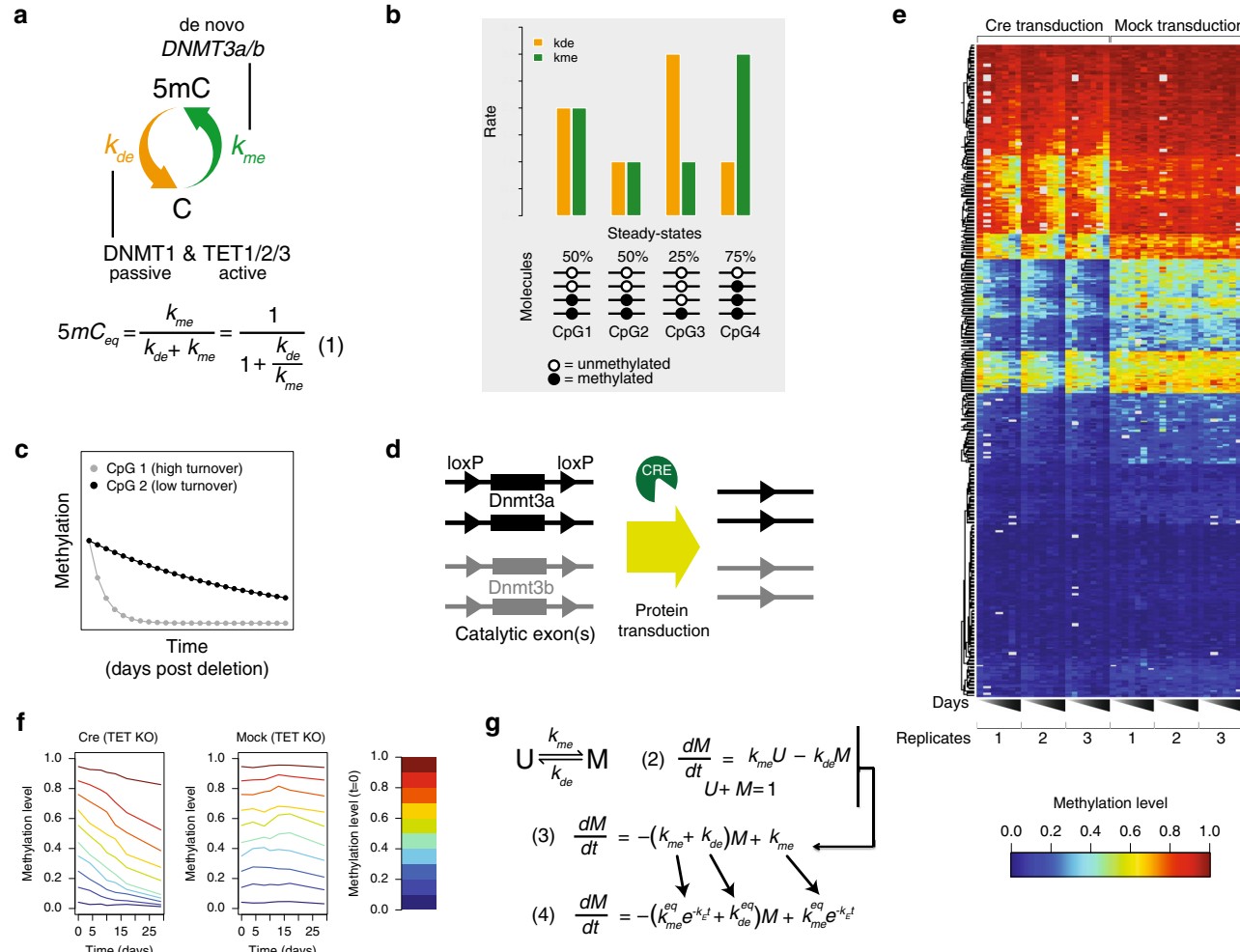

**Fig. 1 A dynamical model and cellular system to infer methylation and demethylation rates. a** Graphic representation of methylation and demethylation rates. The orange and green arrows represent $k_{de}$ and $k_{me}$, respectively. The enzymes responsible for influencing rates are noted. The ratio of rates determines overall methylation levels (Equation (1) below). **b** Example steady-state methylation levels resulting from different $k_{me}$ (green) and $k_{de}$ (orange) combinations. Higher methylation levels are established when $k_{me}$ is larger than $k_{de}$, while low methylation levels represent the opposite. CpGs with the same steady state can have different rates as shown here for 50%. **c** Theoretical trace of methylation loss over time post Cre transduction for two CpGs with similar steady states. **d** Cellular system for genetic ablation of $k_{me}$. Dnmt3a and Dnmt3b with loxP sites flanking catalytic exons. Cre protein transduction allows for efficient genetic deletion of all four alleles. **e** Heatmap of methylation levels for 405 CpGs as measured by amplicon bisulfite sequencing. The left half represents methylation levels for triplicate experiments measured 0, 4, 8, 10, 13, 17, and 29 days post Cre transduction. The right half represents triplicates for mock-treated samples. **f** CpGs were binned based on starting methylation in 10% increments, and the mean decay over triplicates for Cre-transduced samples (left) and mock samples (right) are shown. **g** Dynamical model for DNA methylation and implementation of the exponential dampening factor $k_e$ for affecting $k_{me}$ over time (Eqs. (2)–(4)). See text and "Methods" for details.

limitation, we opted for protein transduction, directly adding Cre protein to cells in culture. This system takes advantage of an engineered Cre recombinase that enters the nucleus via a lipophilic tag at the N-terminus[47]. This proved highly efficient, as nearly 90% of alleles for both enzymes were removed (Supplementary Fig. 1e). While the genotypic proportions of intact alleles will differ between cells, both DNMT3a and DNMT3b signal were undetectable by western blot post Cre transduction (Supplementary Fig. 1f), suggesting most of these functional enzymes were removed. Transcript and most importantly protein levels of DNMT1 and UHRF1 remained comparable upon loss of *Tet1/2/3* and *Dnmt3a/b*, arguing that the maintenance machinery is intact in these genetic backgrounds (Supplemental Fig. 1g, h).

Using the TTKO line, we measured DNA methylation 0, 4, 8, 10, 13, 17, and 29 days post *Dnmt3a/3b* deletion and focused initially on selected genomic sites with amplicon bisulfite

sequencing (Fig. 1e, f, see "Methods"). For this, we assayed 88 genomic regions with disparate steady-state methylation levels, representing TF binding sites, as well as fully methylated regions and promoters (Supplementary Table 1, see GEO submission). Across the time course, methylation levels declined reflecting the absence of de novo methylation activity. Loss of methylation was reproducible (Fig. 1e) and the deep coverage enabled us to analyze 405 CpGs in detail (Supplementary Table 3, see GEO submission and methods for filtering). These data indeed reveal that methylation decays over the time course of the experiment, demonstrating that rate assignments should be possible in this system.

In order to infer de novo methylation and passive demethylation rates from the data, we devised a dynamical model for DNA methylation that mimics the loss of DNMT3 over time (see "Methods" for detailed description). Starting from a framework used previously to describe DNA methylation dynamics[43], we

modified the rate equation to simulate exponential loss of DNMT3 over the course of the experiment. We achieved this by including an exponential dampening factor for $k_{me}$ with the reasoning that the $k_{me}$ gradually decreases after genetic deletion (Fig. 1g). This is an essential consideration, because *Dnmt3* alleles are not deleted instantaneously and protein/RNA are lost as a function of time. This effect can be observed by inspection of the raw data. Instead of following an exponential decay, an attenuation in methylation loss can be seen at the beginning of the time-course experiment (Fig. 1f). The rate of DNMT3 loss is governed by one single parameter $k_e$ that we set to reflect various aspects of the cellular system. Given that average protein half-life has a median of 46 h[48] and doubling time is 16 h in mESCs[49], dilution via cell division is likely the largest contributing factor to DNMT3 loss. If the RNA and *Dnmt3* alleles would disappear instantaneously, the DNMT3 loss over time would occur at a maximum rate of $\log(2)/(16/24) = 1.03$, in units of days. Because neither RNA nor genetic loss of Dnmt3 is an instantaneous process, we set $k_e$ to half of the theoretical maximum rate. Taken together, we designed a theoretical and experimental system to infer methylation and demethylation rates in the genome that is reproducible and can account for methylation patterns observed in ESCs.

**Rate inference reveals the identifiable landscape**. Any measure of decay kinetics is limited by detection accuracy and temporal resolution. For example, it is intrinsically more difficult to capture extremely fast kinetics or a decay that starts at very low signal intensity (i.e., low methylation levels). We thus devised a rate inference method that allowed us to both fit rates optimally and determine the confidence at which they can be determined. To this end, we coupled the dynamical model for DNA methylation to a statistical error model (Supplementary Fig. 2a). We then used Bayesian statistics to calculate maximum likelihood estimates, as well as credible intervals for the rates (see "Methods").

One advantage of this rate inference strategy is that it can provide a complete picture of the assay's detection limits. For any given combination of $k_{me}$ and $k_{de}$, we can determine our ability to infer either $k_{me}$ or $k_{de}$. Applying this to all possible rate combinations resulted in the identifiable landscape (Fig. 2a). Overall, $k_{me}$ is more difficult to infer than $k_{de}$ because the latter is closer to what we are actually measuring. The resulting identifiable landscape revealed a central area of high-confidence parameter estimation (Fig. 2b, case 1), and three areas where rate inference is more difficult (Fig. 2b, cases 2–4). As expected, extreme combinations are hard to retrieve. The most difficult is case 2, representing unmethylated CpGs (high $k_{de}$ and low $k_{me}$). These CpGs are unmethylated throughout the time course and thus provide no information. Additionally, 50% methylated cytosines with very low rates would be difficult to infer (Fig. 2b, case 3), as very little decay is observed throughout the time-course coupled with the higher variability in measuring 50% methylated cytosines. Perhaps least intuitively, in the case of highly methylated cytosines, only $k_{me}$ was difficult to infer (Fig. 2b, case 4). While decay ($k_{de}$) is easily quantifiable at high methylation levels, hitting the methylation ceiling close to 100% prohibits the exact inference of $k_{me}$.

Many of the cytosines covered in our amplicon dataset revealed rate combinations that we could infer with high confidence (Fig. 2c, blue points). Indeed, many of the problematic inference zones were not represented. For example, CpGs with a steady-state methylation level of 50% but very low turnover (bottom left corner of Fig. 2c, i.e., very low rates) were rare. It is important to note that while these rate combinations are inferred with lower confidence, they still would be detectable within these regions, as

can be seen for CpGs with very low steady-state methylation levels (Fig. 2c, bottom right red and black points). Taken as a whole, our inference method is capable of accurately discriminating regimes where rates can be inferred, and the majority of the cytosines we profiled reside in high-confidence regions.

**Rate inference at the genomic scale**. At first glance, determining methylation and demethylation rates at the genomic scale appears straightforward, as it requires bisulfite sequencing of all samples for the time-course experiment. However, the deep coverage required for proper rate assignments (minimum of 50× coverage for 24 samples) makes this cost prohibitive unless genome complexity is substantially reduced. To accomplish this, we used the SureSelect system that employs RNA baits homologous to 297,000 genomic regions (Fig. 3a, see "Methods"), predesigned to enrich for regulatory regions and disease relevant loci. This enrichment was apparent when inspecting raw reads (Fig. 3a) and we confirmed the observation at the global level, with nearly 90% of all mapped reads localizing within 200 bp of bait regions (Fig. 3b). In total, this resulted in a mean of 234 million reads mapped per library, sufficient coverage for high-confidence rate inference at 860,406 CpGs, representing 151k unique genomic locations (~51% of SureSelect baits) and ~4% of CpGs in the mouse genome (Supplementary Table 4, see GEO submission).

Methylation levels decayed steadily over time after *Dnmt3* deletion (Supplementary Fig. 3a). Variance in methylation measurements could be predominantly explained by random sampling of reads from a binomial distribution, whereby variance in methylation levels for cytosines scaled with coverage as expected from random sampling[50] (Supplementary Fig. 3b). Patterns of methylation across CpGs measured at the given sampling time points were highly reproducible across replicates with clustering driven predominantly by the time point analyzed (Fig. 3c). Additionally, for CpGs quantified using both amplicon sequencing and SureSelect (259 of 405), methylation levels were well correlated ($R = 0.98$, Supplementary Fig. 3c). Demethylation rates for ~40% of the cytosines could be assigned with high confidence (860k of $2.1 \times 10^6$), revealing that passive demethylation rates vary widely. On average demethylation rates show an interquartile range of 1.7 fold, but CpG demethylation rates can vary up to 158 fold, exposing that CpGs sharing the same steady-state methylation levels can have highly different kinetics of methylation loss (Fig. 3d, right). Importantly, while a significant proportion of the probe design represents active regulatory elements, ~40% of regions we assay with SureSelect lack DHS signal in ESCs (Supplementary Fig. 3d). Thus, CpGs are represented in active regulatory regions, as well as inaccessible intergenic domains. In summary, we reproducibly inferred rates of methylation and demethylation for nearly 1 million CpGs in mouse ESCs. By observing different kinetics for shared steady states, our data uncover the dynamic aspect of the methylome normally masked in steady-state measurements.

**Rate combinations reveal context-specific activity**. Having extended our rate inference to the genomic level, we first interrogated the relationship between rates of de novo methylation and passive demethylation (Fig. 4a). The relationship between these rates is complex, suggesting that genomic context can have different implications for de novo and maintenance methylation. As a reference, steady-state methylation levels are depicted in Fig. 4a by 45° lines, as the ratio between rates is constant along each line.

Upon closer inspection of the rate relationship, two particular patterns emerge. First, CpGs with a steady-state methylation above 70% vary greatly in turnover (Fig. 4a, upper arm, points in

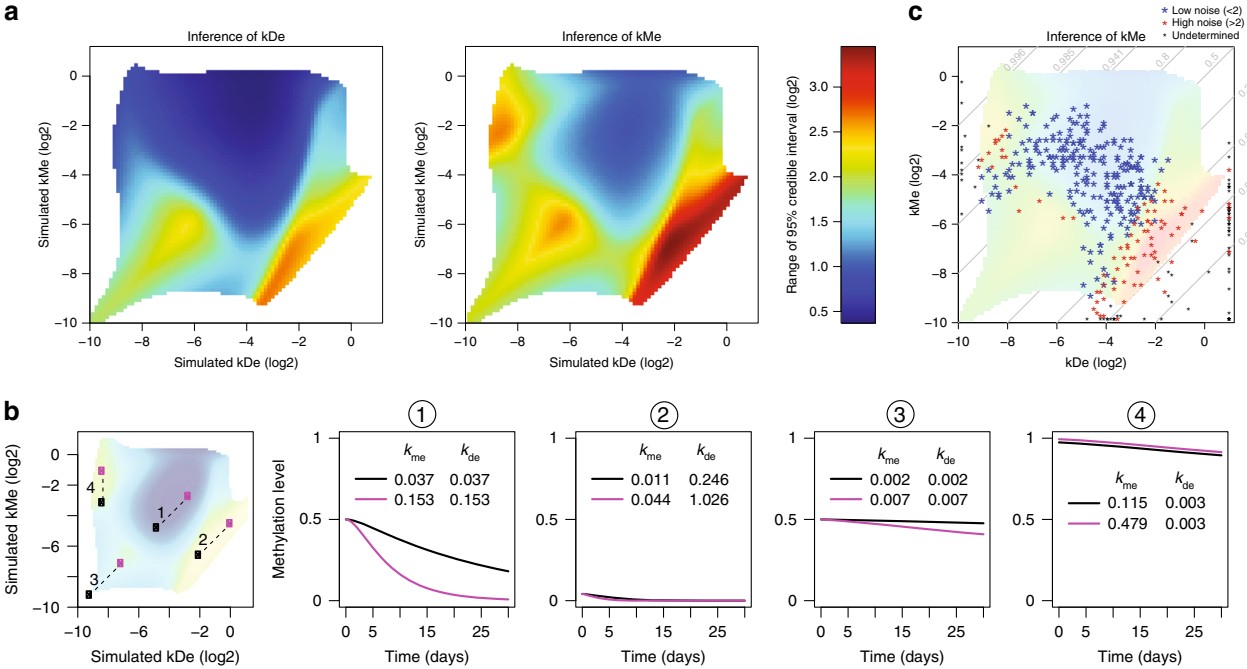

**Fig. 2 The inference landscape for methylation and demethylation rates. a** Inference landscape for $k_{de}$ (left) and $k_{me}$ (right), respectively, given all 6400 possible combinations tested (see "Methods"). Blue regions represent high-confidence regimes were rates can be accurately inferred, whereas CpGs lying in the green to yellow regimes are increasingly difficult and ultimately impossible to determine with high confidence. Confidence levels were determined by an error model explicitly detailed in "Methods". **b** Example pairs of CpGs and their placement in the inference landscape. Theoretical decay curves for the point pairs (connected by dashed lines) are shown to the right. Each pair number trajectory from the far left panel is shown individually in the right four panels. Note, rates for some CpGs can be accurately distinguished (1), while others have reduced confidence (3) or are governed by rates that are not possible to determine (2 and 4). **c** Points representing rate combinations for all CpGs (405) measured with amplicon sequencing. Points are overlaid on the inference landscape taking both $k_{de}$ and $k_{me}$ into account, inference colors are as **a**. Blue points have low noise in rate inference, while red points represent CpGs where noise is high. Black points represent CpGs where rates cannot be determined. Because the logarithm of the rates is displayed on both axes, lines with a slope of one (cases 1–3) correspond to rate combinations that result in the same steady-state methylation level but with different turnover.

the upper left above the 70% methylation line). This suggests a positive relationship between de novo methylation rate and passive demethylation at highly methylated CpGs (see below). Second, CpGs with average methylation below 50% have elevated passive demethylation coupled with variable de novo methylation rates (Fig. 4a, lower arm, points to the lower right of the 50% methylation line). This suggests that low methylation levels observed in genomic methylation maps are a result of both an increase in passive demethylation, as well as a reduction in de novo methylation. Importantly, these relationships remain when accounting for inference confidence levels (Supplementary Fig. 4a). Additionally, regions with higher turnover tended to reside in earlier replicating regions of the genome (Supplementary Fig. 4b) in agreement with their euchromatic context[51].

Next, we asked if these local differences in enzymatic activities could reflect other features of the epigenome. Using a published classifier[52,53] of chromatin states in mouse ESCs[53,54], we assigned each CpG to a particular genomic context defined by TF occupancy and combinations of histone modifications (Fig. 4b). This revealed that particular genomic environments overlapped with specific rate regimes. For example, CpGs outside of genes and regulatory regions (H3K9me3+) show high steady-state methylation, yet surprisingly variable rate combinations. While also highly methylated, CpGs residing in active gene bodies (H3K36me3+) show higher turnover. In contrast, highly methylated intergenic CpGs tend to have reduced passive demethylation and de novo methylation rates, revealing higher DNMT1 fidelity at these regions coupled with reduced DNMT3 activity.

CpGs within active regulatory elements and at insulator regions (CTCF+), in contrast, have reduced overall methylation levels as shown previously[5], but reveal an intriguing relationship in regards to total activity. Passive demethylation is elevated indicating reduced maintenance by DNMT1 at active regulatory elements, including strong enhancers (NANOG+, OCT4+, H3K27Ac+, H3K9Ac+, and H3K4me1/3+). At the same time, de novo methylation activity varies widely in these regions, potentially indicating that transacting factors present at specific regulatory regions may affect DNMT3 activity differently (see below). While many promoter CpGs (H3K4me3+, H3K27Ac+, and H3K9Ac+) are also present in this regime, their rates cannot be determined with high confidence as their steady-state methylation levels are too low to allow for robust kinetic analysis (Supplementary Fig. 4a). In summary, our results reveal that methylated cytosines have both highly variable methylation kinetics and a surprising abundance of sites with high rates of passive demethylation that is normally masked by de novo methylation activity.

### Site specificity of active demethylation by the TET enzymes.
Having quantified passive demethylation at the genome scale, we sought to determine the contribution of TET-dependent, "active" demethylation at these CpGs. More specifically, we sought to interrogate how TET proteins affect demethylation rates at the CpGs we measured previously. Our model framework would predict that we can determine the change in demethylation rates by comparing steady-state methylation levels between ESCs with and without TET proteins (Supplementary Fig. 4c). We therefore

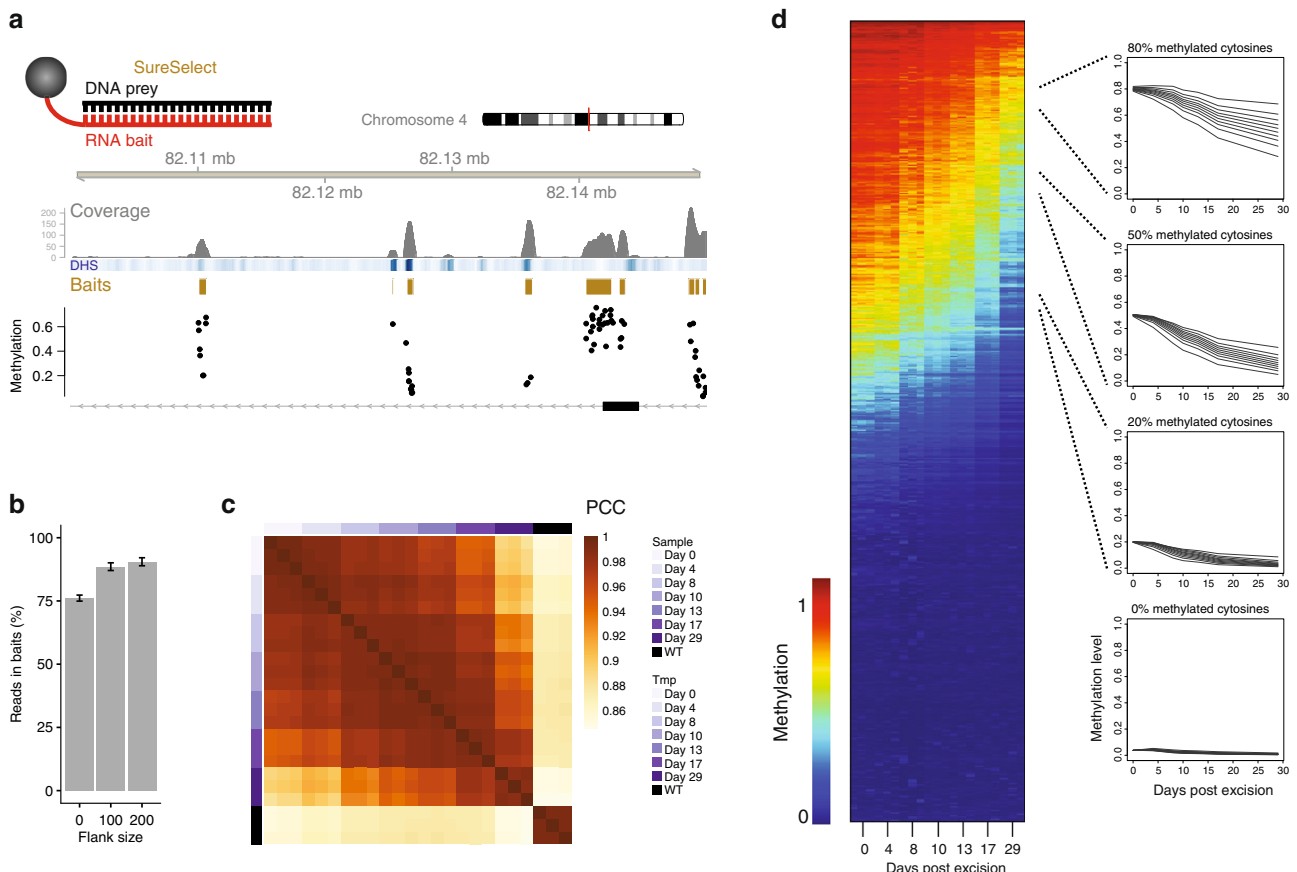

**Fig. 3 Genome-scale measurement of methylation kinetics. a** Outline of the SureSelect strategy (above) and a browser screenshot of raw read counts, DHS signal (blue), bait design regions (gold), and CpG methylation level measured prior to induced deletion (black dots). **b** Percentage of reads in libraries mapping to bait regions. Bars represent bait region boundary extension by 0, 100, or 200 bp, respectively. Error bars represent two standard deviations from the mean of three replicates. **c** Hierarchical clustering of methylation levels for all samples measured. Annotation column and row depicts days post transduction and wild-type samples. PCC is the Pearson correlation coefficient. **d** Decay of methylation over time for 2.1 million CpGs. Color scale is as in Fig. 1e, dark red representing 100% methylated CpGs and dark blue representing 0% methylation. Traces to the right represent average profiles of CpGs with similar steady states (noted above panels) but different decay kinetics. CpGs with steady states of 20, 50, and 80% (±2%) were separated into decile bins based on $k_{de}$ and average profiles from these bins are shown.

measured methylation levels at these CpGs in the presence of TET1/2/3 and observed an almost unidirectional increase in methylation levels when TET proteins are absent (Fig. 4c). This observation is indeed in agreement with the defined role of TETs as demethylases, and our model assignment of these proteins as demethylases. In this context, we refer to the contribution of TETs to the demethylation rate as TET activity. More specifically, this number represents the fold change in $k_{de}$ when TETs are present (in log2 space). To validate rate predictions in an independent manner, we performed amplicon bisulfite time-course experiments following *Dnmt3* deletion in the parental cell line with TET activity (Supplementary Table 2). As expected, $k_{de}$ values determined by the time course showed very high correlation ($R = 0.88$) to those predicted using only changes in steady-state methylation (see Supplementary Fig. 4c–h and "Methods"), representing a strong validation of our modeling strategy.

The presence of TET activity drastically affected demethylation rates throughout the genome, increasing them at least threefold for half of all CpGs analyzed (~476,895 CpGs, Fig. 4c inset). This effect is most apparent at enhancer elements, followed by polycomb marked regions and gene bodies (Fig. 4d). Taken together, this revealed that TET proteins have a considerable influence on the demethylation rate. This contribution in ESCs

tends to be greater than DNMT1 infidelity and is highest at active distal regulatory elements.

Next, we asked whether active and passive demethylation scale in a similar fashion. While TET activity is highest in regions with elevated passive demethylation (Supplementary Fig. 4i), the relationship is complex, as rates of passive and active demethylation vary widely between individual CpGs (Supplementary Fig. 4j). While TET activity reaches its maximum in active regulatory elements and bivalent domains, regions with the highest levels of passive demethylation reveal little TET activity. Several of these CpGs indeed overlap closely with TF binding sites as can be seen for CTCF (Fig. 4b, Supplementary Fig. 4i, j), suggesting that continuous presence of TFs inhibits both TET and DNMT1 activity directly at the site of binding. In contrast, heterochromatic intergenic regions generally have both low active and passive demethylation. We conclude that CpGs reside in different rate regimes as a function of genomic context, and TET activity has an overall effect of increasing demethylation rates throughout the genome, but particularly at enhancers.

**Transcription correlates with methylation turnover.** CpGs with particularly high steady-state methylation levels (≥70%) displayed

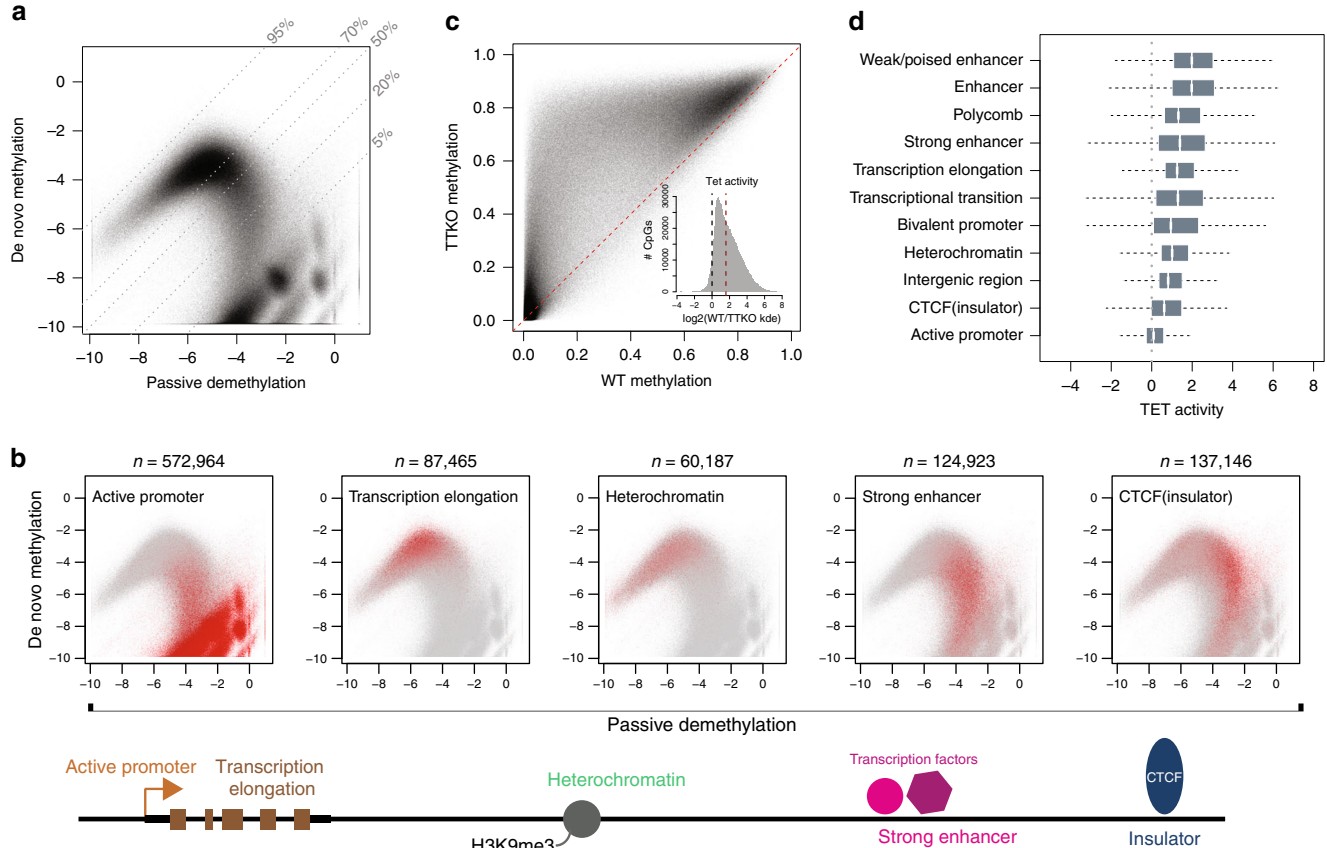

**Fig. 4 Rate combinations are characteristic of particular genomic contexts. a** Scatterplot of $k_{de}$ and $k_{me}$ for all cytosines. Dashed lines represent different steady-state methylation levels, which are noted on the right upper borders. **b** Scatterplots as in **a**, but with CpGs colored according to overlap with previous genome annotations[52,54]. Red points represent CpGs of interest for the particular genomic annotation. For example, in the first panel all CpGs overlapping with promoter regions are shown in red, while all CpGs outside of promoters are shown in gray. The number of CpGs overlapping with each state are noted above the respective panel. A graphical depiction of the genomic annotations for the five different contexts is shown below the scatterplots. **c** Scatterplot of methylation levels measured in wild-type (x-axis) and TTKO cells (y-axis). Inset: change in $k_{de}$ as a function of TET activity. Values on the x-axis represent $log2(k_{de}^{WT}) − log2(k_{de}^{TTKO})$. The vertical blue line represents CpGs where TET activity has no effect on $k_{de}$, while the red vertical line represents a three-fold increase in the demethylation rate as a function of TET activity. Note the almost unimodal shift in steady-state methylation levels underscoring the role of TET proteins as demethylases. **d** TET mediated changes in $k_{de}$ as a function of genomic context. TET activity is as defined above in **c**. Annotated regions are sorted based on mean change in $k_{de}$. The box represents the middle 50% of the data, the line inside the box is the median, and whiskers are defined by the most extreme values lying within 1.5 times the interquartile range.

a remarkable linear relationship between de novo methylation and passive demethylation rates (Fig. 5a, red points). We reasoned that CpGs with high steady-state methylation but different rates of turnover might be residing in regions of different transcriptional activity (Fig. 4b). To address this, we grouped genes based on their transcriptional output and tallied steady-state methylation, de novo methylation rate, and active/passive demethylation rate as a function of relative position in the gene (Fig. 5b, Supplementary Fig. 5a). This revealed that total methylation turnover increases with transcriptional activity and in turn argues that the high overall methylation observed at genes is in constant flux as a function of transcription. This is also evident for individual rates as methylation by DNMT3 increases with transcriptional output (Fig. 5b). Recruitment of de novo activity in genes likely involves H3 methylation at lysine 36. The presence of this modification increases with transcriptional rate and it is recognized by DNMT3b[55], which has been suggested to be functionally required for genic methylation[56].

The requirement for continuous de novo methylation may arise due to higher demethylation rates at transcribed genes. In addition to de novo methylation rate, both active and passive demethylation increase in gene bodies with transcriptional

output, although to a lesser degree. Importantly, however, measured turnover rates are largely independent of whether CpGs reside in introns or exons (Supplementary Fig. 5b). Furthermore, this signal is unlikely to result from increased accessibility in transcribed gene bodies, as we do not observe a higher prevalence of DNAseI hypersensitive sites in highly transcribed gene bodies (Supplementary Fig. 5c). However, we do observe increased histone turnover as revealed by H3.3 ChIP signal[57] (Supplementary Fig. 5c). This links transcription coupled deposition of replication-independent histones[58] with reduced DNMT1 fidelity. In summary, transcription coincides with high turnover of DNA methylation at genic regions.

**Heterochromatin and euchromatin show opposing turnover rates.** CpGs with high steady-state methylation but variable turnover also exist outside of genic regions, allowing us to explore their relationship with other chromatin marks. This revealed that CpGs with high methylation but low turnover were progressively enriched for the heterochromatic marks H3K9me2 and H3K9me3 (Fig. 5c, Supplementary Fig. 5d, e). This positive correlation between DNMT1 fidelity and presence of H3K9

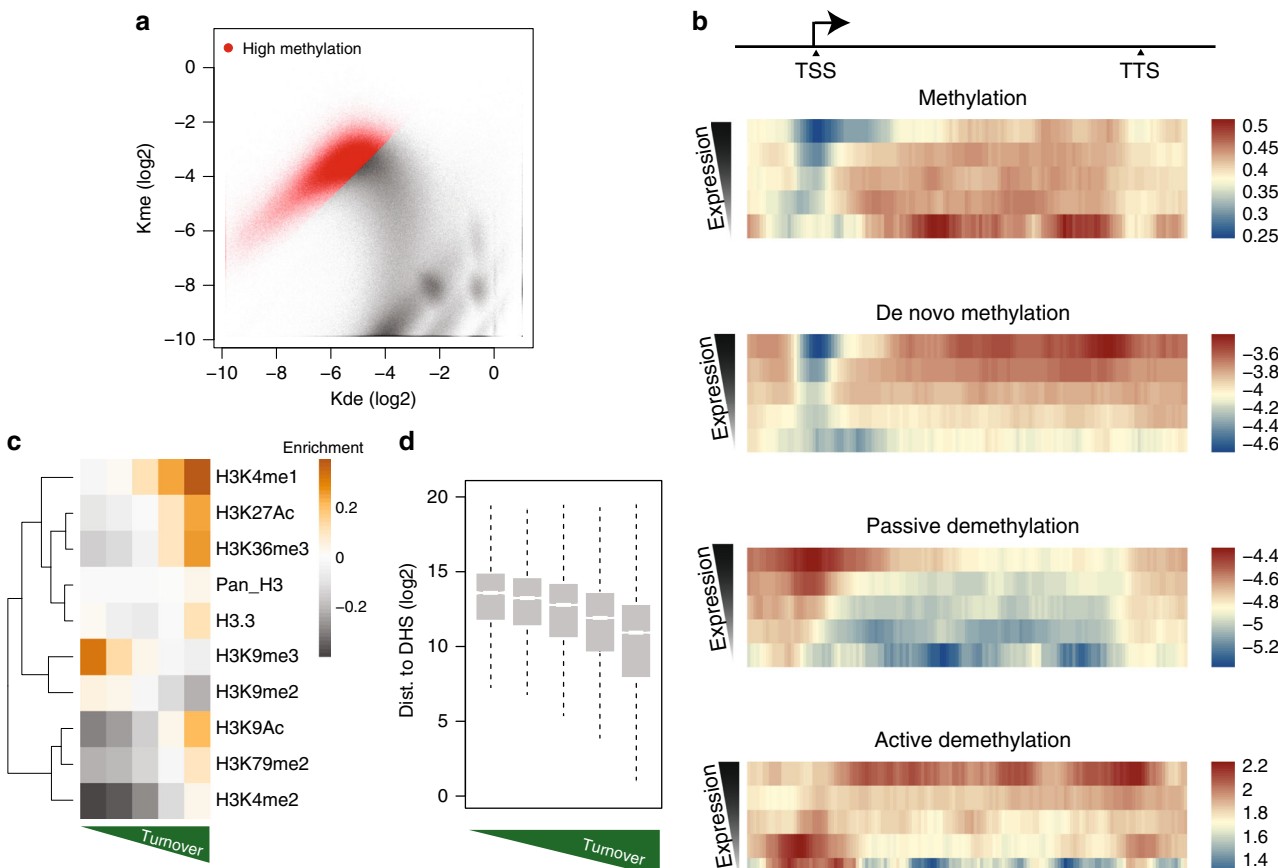

**Fig. 5 Turnover at highly methylated cytosines correlates with genomic activity. a** Scatterplot of $k_{de}$ and $k_{me}$ highlighting CpGs in red that have a high steady state ≥70%). **b** Rates and methylation levels as a function of location in genic regions. Mean values for CpGs are represented as a function of their position in genes as a percentage (i.e., each genic region represents 100 bins). Upstream and downstream of noted TSS and TTS regions represent 10 kb of flanking DNA. Each row in the heatmap represents a collection of genes binned on transcriptional output in RPKM (five bins total), with the highest expressing bin on top. Each bin represents at least 2k genes. **c** Heatmap representing signal for eight different chromatin marks across bins of highly methylated cytosines (red points from **a**). Mean histone signal was calculated by tiling the genome into 1 kb bins and determining enrichment in ChIP signal over input for the respective marks. From left to right, bins are split based on mean methylation turnover within the bin, with the highest turnover bin on the far right. Note the increase in H3K36me3 and active marks, with the concomitant decrease in H3K9me2/3. **d** Turnover increases with proximity to distal regulatory elements. CpGs were binned on turnover as for **c**, but their distance to the nearest DHS site was calculated. Boxplot elements are as defined in Fig. 4d.

methylation marks is in line with the known interaction of methylated H3K9 with UHRF1[59], an accessory factor required for maintenance methylation[60,61].

Interestingly, a subset of highly methylated cytosines display both high turnover and low H3K36me3. As these CpGs overlap with regions of local enrichment for H3K4me1 and H3K27ac (Fig. 5c), we reasoned that these particular CpGs might be positioned proximal to active regulatory regions despite being hypermethylated. Indeed this is the case (Fig. 5d), as CpGs that border regulatory regions are under a regime of elevated methylation turnover. It has been shown previously that CpGs residing in the proximity of CpG island shores can exhibit variable methylation levels[62,63]. Our data now argue that these CpGs are under a regime of higher turnover even in a cell state, where they are highly methylated. We conclude that hypermethylated CpGs can undergo high methylation turnover, when proximal to regulatory regions or positioned within highly active genes.

**Transcription factor-specific effects on methylation kinetics.** Transcription factor binding coincides with reduced DNA methylation levels observed at regulatory regions, such as enhancers and CpG islands[5]. To ask how methylation turnover

relates to TF presence, we used DNAseI accessibility as a surrogate for TF binding. This revealed that de novo methylation globally decreases with increased accessibility, while both active and passive demethylation increase (Fig. 6a). This shift in rates readily explains the established low methylation levels at *cis*-acting sequences and supports a model where TF binding to regulatory regions reduces DNMT1 and DNMT3 activity, while increasing that of TETs.

To determine if this effect is transcription factor specific, we used publicly available genome-wide binding data for 15 TFs in mouse ESCs and visualized methylation turnover as a function of proximity to bound distal motifs (see "Methods" for ChIP data processing). For several factors, including CTCF, ZC3H11A, and REST, the general reduction in maintenance and de novo methylation was readily apparent surrounding bound sites (Fig. 6b). However, while de novo and maintenance methylation are generally reduced in the vicinity of binding for all factors, discrete patterns were much less apparent for the other 12 TFs. This heterogeneity is likely caused by co-occupancy, particularly in the case of pluripotency factors[64]. In contrast, the unique chromatin structure of both CTCF[65,66] and REST[66] bound sites may enhance rate signatures at bound regions (see below). Of

note, TCFCP2I1 and ESRRB both show subtle patterns of an opposite effect, namely increased de novo and maintenance methylation rates at their motifs (Supplementary Fig. 6b, c). Importantly, this effect is present at sites of binding that are not hypersensitive to DNAseI digestion, suggesting that it is less likely a result of adjacent bound factors. In the case of ESRRB, increased de novo methylation at bound sites seems compatible with the observation that it can bind to methylated enhancers[67,68].

Active demethylation on the other hand followed a more general consensus, namely highest levels of TET activity adjacent to bound sites (Fig. 6b, right). This was evident for all 15 transcription factors analyzed, with OCT4, SOX2, and NANOG revealing up to a 16-fold increase in demethylation rate as a function of TET activity. As the 15 TFs profiled represent members from distinct families, this might indicate that TET proteins are less likely to be specifically targeted through direct interaction, but rather through preferable binding to open chromatin. Indeed TET activity generally increases with increasing accessibility, as suggested by its contribution to the demethylation rate adjacent to bound motifs at distal sites (Fig. 6a right, Fig. 6c). We then reasoned that if these patterns result from TF presence, signal strength should increase as a function of binding. For several factors, including CTCF, ZH3H11A, OCT4, and NANOG, the patterning of rates became more striking with increased ChIP signal (Supplementary Fig. 6a). De novo methylation rates tend to decrease, passive demethylation increases, while active demethylation increases (OCT4 and NANOG) or becomes specifically localized adjacent to the TF in question or in linkers between nucleosomes.

We conclude that regulatory regions show a reduction in de novo and maintenance methylation, and increase in active demethylation as the most prominent pattern. While this is a function of TF binding, some factors reveal different trends suggesting TF-specific influence on these rates.

**Nucleosome occupancy contributes to local turnover**. Having interrogated rates as a function of TF binding, we next asked how methylation turnover changes at highly positioned nucleosomes. In the case of the insulator protein CTCF, bound sites show reduced DNMT1 fidelity and DNMT3 activity compatible with a model of steric hindrance, where TF binding impedes both de novo and maintenance methylation activities (Fig. 6c). Importantly, the region of enhanced passive demethylation included not only the binding site itself, but extended ~250 bp on both sides of the binding site. We reasoned this may be due to highly positioned nucleosomes adjacent to the bound factor. Indeed, using a high-coverage MNase data set that we generated previously[69], highly phased nucleosomes around CTCF sites closely overlap with the region of increased passive demethylation and decreased de novo methylation (Fig. 6c, Supplementary Fig. 6d). Indeed, this pattern includes nucleosomes bordering the binding site and extends until the linker between the first and second nucleosome is reached. These observations suggest that both CTCF binding and bordering nucleosomes reduce DNMT activity. Active demethylation, in contrast, is very low at the motif itself but increases directly adjacent to it and subsequently decreases over the bordering nucleosomes in a fashion similar to DNMT3. Also apparent is that both active demethylation and de novo methylation increase in linkers between nucleosomes and immediately adjacent to CTCF. Taken together, our findings suggest that both factor binding and positioned nucleosomes inhibit DNMT1 and DNMT3, while accessibility is a strong determinant for active demethylation (Fig. 6d). These activities, in turn, likely account for the complex and cell type-specific patterns of reduced methylation levels observed at regulatory regions.

## Discussion

Here, we established a theoretical and experimental framework to quantify local methylation and demethylation activity at single-CpG resolution throughout the genome of mouse ESCs. Studying methylation as a continuous process reveals that methylation levels do not predict methylation turnover, which can differ over two orders of magnitude. This finding was made possible by generating inducible deletions of both de novo methyltransferases in a cellular background, where we removed all three TET enzymes. Quantification of the methylation kinetics in this Penta-knockout over time at high coverage enabled us to infer actual rates of activity at individual CpGs. It revealed that de novo methylation, as well as passive and active demethylation activities are affected by local variations in chromatin, transcriptional activity, and TF binding, leading to complex rate patterns that readily explain steady-state methylation levels.

Our study builds on and extends previous conceptual and empirical attempts[34,37,38] at quantifying methylation activity in different genomic contexts. However, our approach distinguishes itself in several aspects. First, fitting rate combinations using the dynamical model coupled to an error model as a framework to infer activity is, to our knowledge, the first of its kind. One major advantage of this modeling approach is that it allows us to resolve which rate combinations can be inferred in this system. While the inability to fit rates for CpGs at exceedingly low methylation levels is obvious, fitting rates for very highly methylated cytosines can also be problematic. Intermediately methylated CpGs with low rates are likewise difficult to infer, due in large part to the variance in methylation measurements for CpGs approaching 50% steady state. Second, we have determined rates with high confidence at just under 1 million CpGs across the genome enabling high resolution and comprehensive analysis of methylation kinetics.

The rate patterns for de novo methylation we observe are fully compatible with the described inhibition of DNMT3 by histone H3 methylated at lysine 4[70,71]. This could at least in part explain reduced activity of DNMT3 at active promoters and enhancers. Additionally, the increase in de novo methylation rates in highly transcribed gene bodies supports previous findings regarding DNMT3 affinity to H3K36 methylation[55,72], a mark that occurs at transcribed genes and scales with transcription through association with elongating RNA polymerase[73,74]. This de novo activity is required to keep these sequences methylated as it coincides with reduced DNMT1 maintenance and increased TET activity, leading to elevated turnover that scales with transcription rate. Our observation that H3.3 signal scales in a similar manner link histone turnover with reduced DNMT1 fidelity and increased TET activity. Nevertheless, targeted gene body methylation is both ancient, spanning 900 million years of metazoan evolution[75,76], and poorly understood. Several hypothesis have been put forth[77,78], including gene silencing in plants[79], suppression of spurious transcription start sites[56], or a mere byproduct of transposon silencing[76]. While the function of gene body methylation remains mysterious, our observation of increased turnover of methylation in these regions coupled with conservation further argues for a functional role.

Methylation kinetics of highly methylated intergenic CpGs represent another intriguing case. At the global scale, CpGs in this context seem to underlie two regimes. First, CpGs distal from active regulatory elements reside in neighborhoods of increasing H3K9me2/3 and high fidelity of DNMT1. This is in agreement with the observation that the cofactor Uhrf1 recognizes this mark[80–83], and is involved in maintenance activity of DNMT1[60,61]. The second regime represents CpGs increasingly closer to active regulatory elements, which show high turnover driven by both active and passive demethylation. It is tempting to speculate that these

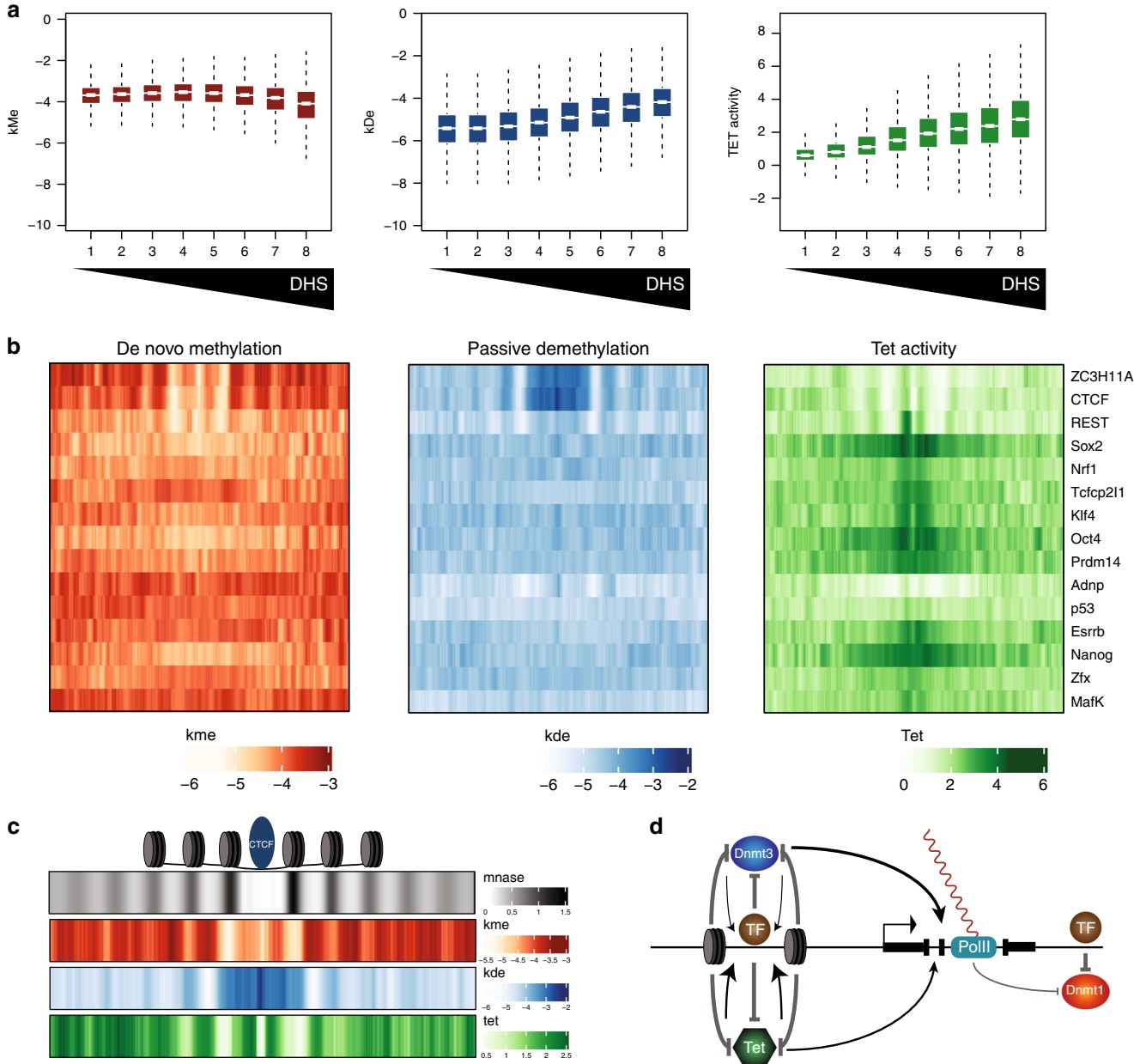

**Fig. 6 Transcription factor binding shows variable effects on methylation and demethylation activity. a** Rates and TET activity as a function of distal DHS signal. The mouse genome was split into 500 bp bins, and reads tallied for all bins that were completely mappable. Bins were then selected as having a minimum distance of 10 kb from an annotated promoter, and split based on number of DHS reads overlapping these bins. DHS signal increases with increasing bin number, where it is apparent that while $k_{me}$ (left) decreases with increasing accessibility, both $k_{de}$ (middle) and TET activity (right) increase. Boxplot elements are as defined in Fig. 4d. **b** Rates and TET activity as a function of distance to bound TF motifs. ENCODE ChIP data for 15 TFs was quantified by counting reads surrounding motifs for each TF in a 201 bp window centered on the motif. Each row of the heatmap represents mean rates as a function of distance to the center of the motif for the respective factor. Sites represented here were selected as the top 900 enriched motif occurrences for each factor (see "Methods" for enrichment determination). **c** Nucleosome positioning, rate of de novo methylation, passive demethylation, and TET activity around bound CTCF sites, color as in **b**. MNase read counts were shifted by 75 bp to reflect position of the nucleosome dyad. **d** Model representing the effect of chromatin processes on methylation and demethylation rates. Presence of bound transcription factors can inhibit both processes, while transcription through gene bodies results in increased de novo methylation and passive demethylation. TET proteins in contrast tend to illicit the strongest effect on demethylation rates at accessible regions proximal to bound transcription factors.

methylation dynamics create binding opportunities for methylation sensitive transcription factors[84–86].

Time-course measurements in the presence and absence of TET proteins allowed us to distinguish between active and passive demethylation. One surprising result is the reduced fidelity of DNMT1 around distal regulatory regions (see below). Global

demethylation had been initially observed in long-term cultures of DNMT3 knockout stem cell lines[28]. While this loss was attributed to DNMT1 infidelity, it is important to note that TETs were present in these cells, but had not been discovered at the time this study was published. Indeed, we show that TET proteins have a significant effect on demethylation rates at hundreds of

thousands of CpGs scattered across the genome, and that TET activity has a stronger effect on the demethylation rate than DNMT1 infidelity. It is important to note that we cannot distinguish whether the effect of TET activity on demethylation rate is due to bona fide active demethylation (base excision), or incomplete maintenance of oxidized methyl groups[87].

Complete loss of TET activity caused an almost unimodal increase in steady-state methylation, which is partially at odds with previous findings in TET mutants where hyper- and hypomethylated regions were more evenly represented[18,88]. While this discrepancy could in part be due to our enrichment of active regulatory elements, approximately half of the bait regions in our study do not overlap with DHS signal. Our observation of a nearly exclusive hypermethylation phenotype is in support of a role for this protein in increasing the demethylation rate, as we have assigned in our model. While the effect of reduced methylation by TET enzymes on gene regulation is not completely understood, TET function has been implicated in activation of differentiation specific genes[18]. Our observation that TET activity is generally enhanced at bound TF sites raises the intriguing possibility that the permissive chromatin environment afforded by TF binding can help to create regions of local hypomethylation[89–92]. In turn, this local hypomethylation could create an opportunity for the binding of additional TFs. Moreover, the scaling of multiply oxidized bases with local chromatin accessibility has been previously documented[93], and thus we speculate that TET activity is less likely caused by specific recruitment but rather accessibility caused by factor binding. While we cannot rule out the possibility of recruitment as has been documented for selected factors[94–97], our data are more in support of the simple scenario where TET proteins have higher activity at accessible regions. Our data further argue that TF binding inhibits both DNMT3 and DNMT1 activity, producing unmethylated cytosines that can be recognized by the CXXC domains of TET proteins. Local recruitment of TET proteins could then serve to increase TET activity on neighboring methylated cytosines. In the case of TET1 and TET3, this interaction can be facilitated directly by their CXXC domains[98–100], and in the case of TET2 possibly through its interaction with IDAX[101]. The preference for TET activity at accessible sites is further supported by our observation of increased activity in linker regions of positioned nucleosomes, with decreasing activity over the nucleosome itself.

Indeed, using ChIP-Seq data from 15 transcription factors, we find that maximum TET activity is localized in the immediate vicinity of transcription factor bound sites, where higher accessibility is expected. This seems to be partially independent of steady-state methylation levels, as high turnover is also prevalent at CpGs with high steady-state methylation levels nearby and flanking DHS sites. We interpret this to be the net result of increased DNMT1 infidelity, de novo methylation rate, and TET activity. The net result is a CpG site with higher methylation levels and elevated turnover, and these CpGs tend to occupy the borders of regulatory elements.

CpGs located at active regulatory regions reveal highly variable kinetics. One of the most striking is the elevated maintenance error coupled with the more variable rate of de novo activity. Both these observations fit a model whereby steric hindrance results in reduced methylation (Fig. 6d). Indeed, it has been shown that active regulatory elements reveal slower kinetics of remethylation after passage of the replication fork[102]. These results are compatible with a model, where many DNA binding factors rebind their consensus motifs quickly after replication and in turn interfere with the maintenance methylation reaction.

Similarly, factor occupancy in other phases of the cell cycle could inhibit de novo methylation activity. Reduction in de novo activity

and maintenance fidelity increase with local accessibility, and taken together contribute to low steady-state methylation levels observed at regulatory regions. This scenario creates ample molecular opportunities for TFs to create a region of reduced methylation as we have shown previously for REST[5], which could enable binding of DNA methylation sensitive TFs, such as Nrf1[103]. Surprisingly, although several of the 15 tested TFs seem to cause reduced methylation, there are notable exceptions. For example, ESRRB and TCFCP2l1 sites show a slightly different pattern of rates at distal bound sites with low DHS signal, namely an increase in both de novo and maintenance activity. Both factors are specific to ESCs and may serve as early binding proteins in the hierarchy of pluripotency factor enhancer binding[67].

Taken together our combination of theoretical and experimental work reveals a significant layer of information previously unresolved by methylome profiling. It exposes this part of the epigenome as a highly dynamic entity within particular genomic contexts.

## Methods

**Cell line generation.** ESC lines conditionally deficient for Dnmt3a and Dnmt3b were derived by outgrowth of blastocyst embryos obtained by crossing mice doubly homozygous for floxed alleles of Dnmt3a and Dnmt3b[46]. The mouse strain was maintained on a C57BL6/J background. Mice were genotyped by PCR and ESCs derived from a homozygous clone. The Dnmt3a/3b flox/flox line was passaged on feeder cells. TET TKO cells were generated from this line using a previously described protocol[44]. In short, guides directed at the catalytic exons of the TET enzymes were cloned into the pX330 vector and all three were cotransfected into the Dnmt3a/3b flox/flox cell line. DNA was extracted from individually picked clones and PCR product amplified overlapping the CRISPR cut site. The PCR fragment from clones was then treated with a restriction enzyme whose recognition sequence is close to the cut site, thus undigested fragments would represent mutated alleles. Alleles were sequenced from one clone displaying likely mutations in all six alleles (Supplementary Fig. 1).

**Slot blot.** Slot blotting of putative TTKO clones was carried out following an established protocol[104], using an antibody against 5hmC (#39769, Active Motif) and 5mC (BI-MECY-1000, Eurogentec). Genomic DNA was denatured with 4 N NaOH and the solution was neutralized by addition of an equal volume of 2.5 M ice-cold NH$_4$Ac. The single-stranded DNA was spotted on a TE-soaked nylon membrane and then baked at 80 °C for 30 min and UV cross-linked.

**Cre transduction.** ESCs were cultured on feeders[105] and passaged at least once on feeders prior to trypsinization for Cre protein transduction. For transduction[106], ESCs were trypsinized, resuspended in PBS and quantified. Approximately 2.5 × 10$^5$ cells were transferred into fresh falcon tubes, spun down, and resuspended in 500 µl of filtered serum-free medium containing either 1 µM Cre protein or an equivalent volume of Cre dialysis buffer (2 M NaCl, 50 mM HEPES pH7.4, 1 mM DTT, 1 mM EDTA, and 5% Glycerol). The cells were then plated in 24-well plates precoated with feeders (2–48 h in advance) and prewashed twice with PBS. After 16 h, cells were washed twice with PBS and coated with FCS-based ES medium[105]. ESCs were transferred to gelatin-coated feeder-free six-well plates 24 h and to 10 cm plates 72 h after transduction. Pellets were collected from trypsinized cells at indicated time points and culturing was continued until 29 days post transduction in a feeder-free environment. All ESCs used for Cre transduction experiments were cultured for at least ten passages prior to Cre transduction.

**DNA extraction.** Genomic DNA was extracted by resuspending cells in 1% SDS with 50 µg proteinase K and incubation at 55° for 5 h. The cell lysate was then mixed at a 1:1 ratio with a mix of phenol:chloroform, spun at max speed for 5 min, and the upper aqueous layer was retained. A second phenol:chloroform extraction was performed, and subsequently chloroform was added at a 1:1 ratio, mixed, and spun for 5 min at RT at 12,000 × g. The upper phase was retained and DNA precipitated by adjusting the aqueous phase to 300 mM NaOAc and >70% ethanol followed by centrifugation at 12,000 × g at 4 °C. The DNA pellet was washed with 70% ethanol, dried, and resuspended in 10 mM Tris pH 8.0. DNA was treated with 50 µg/ml RNase and precipitated using 300 mM NaOAc and >70% ethanol as above.

**TaqMan genotyping.** Primers and probes were designed complementary to sequences between the loxP sites for Dnmt3a and Dnmt3b (see Supplementary Fig. 1 for sequences). For the reaction, 30 ng of genomic DNA, 900 nM of primers, and 0.25 nM of probe were mixed with 1× TaqMan Universal PCR master mix in a total volume of 25 µl. Cycling conditions were an original incubation of 2 min at

50° followed by 10 min at 95°, and then 40 cycles of 15 s at 95° and extension for 1 min at 60°. Primers and a probe were designed for *Gapdh* to use for normalization, and relative allele frequency was calculated using a previously described method[107]. For comparative purposes, DNA from *Dnmt3a/Dnmt3b* knockout cells was mixed with wild-type ESC DNA at ratios of 100:0, 30:0, and 0:100, respectively.

**Western blot**. Whole cell lysate was extracted by resuspending ~$1 \times 10^6$ cells in 100 μl RIPA buffer (50 mM Tris pH 8.0, 150 mM NaCl, 1% NP-40, 0.5% sodium deoxycholate, and 0.1% SDS) followed by incubation for 30 min at 4 °C. Samples were then sonicated for six cycles of 30 s on, 30 min off on the Diagenode Bioruptor and subsequently spun for 20 min at $10,000 \times g$ while cooled to 4 °C. The supernatant was retained, diluted 50× and quantified using the Micro BCA Protein Assay Kit from Pierce. Approximately 20 μg of protein was separated using a Thermo Fisher NuPage 3–8% tris-acetate gel and transferred onto an activated PVDF membrane. The membrane was blocked for 1 h at room temperature in 5% milk resuspended in TBST (10 mM Tris, 1.5 M NaCl, and 1% Tween-20). Antibodies for DNMT3A (Novus 64B1446, 1:1000), DNMT3B (Imgenex IMG-184A, 1:1000), DNMT1 (AbCam ab188453), UHRF1 (MBL D289-3), or LAMIN B1 (AbCam ab16048, 1:10,000) were diluted in 5% milk in TBST and incubated on the membrane overnight at 4 °C with rotation. Membranes were washed four times for 4 min in TBST, and incubated with HRP-conjugated secondary antibodies (GE RPN4201V for DNMT3A and DNMT3B, GE NA934V for LAMIN B1 and DNMT1, Sigma AP136P for UHRF1, 1:10,000 dilution for all) for 1 h at RT. Membranes were washed four times for 4 min in TBST, incubated for 2 min with Advansta WesternBright Sirius chemiluminescent substrate and acquired on the GE Amersham Imager 680. All signals in Supplementary Fig. 1f are from the same membrane. The membrane was stripped after each individual blot by incubating the membrane for 15 min in Thermo Scientific Restore Western Stripping Buffer (#21059), washed three times for 4 min in TBST followed by blocking and incubation of primary antibodies as described above.

**Amplicon bisulfite sequencing**. Approximately 2 μg of extracted genomic DNA from days 0, 4, 8, 10, 13, 17, and 29 were first mixed with 3.2 pg of both unmethylated Lambda bacteriophage and in vitro methylated T7 bacteriophage DNA. Addition of the bacteriophage DNA was used to control for bisulfite conversion efficiency. Samples were then bisulfite converted using the EpiTect kit from Qiagen per the manufacturer's instructions. Primers designed to amplify UMRs, LMR, and FMR regions (88 in total, Supplementary Table 1, see GEO submission) were distributed in 96-well plates and amplification was carried out using Amplitaq gold and the following thermocycler settings (all temperatures are in Celsius and all incubation times are 30 s unless specified): 1 cycle of 95° for 9 min, 20 cycles of touchdown with 95° melt, 55°–51° annealing, and extension at 72°, followed by 36 cycles of 95° melt, 51° annealing, and 72° extension.

The amplicon reactions were then mixed, run out on an agarose gel, and DNA extracted. Libraries were then constructed using the NEBNext ChIP-seq Library Prep kit (#E6240) as per the manufacturer's instructions, indexed, pooled, and sequenced using the Illumina MiSeq platform in 250 bp paired-end mode. The last 100 base pairs were trimmed due to reduced sequencing quality, and aligned to the mm9 build using Rbowtie in the QuasR package and the following parameters: genome = "BSgenome.Mmusculus.UCSC.mm9", paired = "fr", and bisulfite = "undir".

**SureSelect sequencing**. SureSelect enrichment and subsequent sequencing of bisulfite DNA was carried out as per the manufacturer's instructions. Briefly, genomic DNA isolated from the time points above was sonicated down to 150–200 base pairs in size using a Covaris S220, followed by library construction. The libraries were then hybridized to probes (Mouse Methyl-Seq XT, 931052), bound to streptavidin beads, washed, and bisulfite converted using the EZ DNA methylation Gold kit from Zymo (D5005). Bisulfite-converted DNA libraries were then amplified and indexed, pooled and sequenced in 51 bp paired-end mode using the Illumina HiSeq platform. Sequenced reads were aligned to the mm9 build using Rbowtie in the QuasR environment with the following parameters: genome = "BSgenome.Mmusculus.UCSC.mm9", bisulfite = "dir", aligner = "Rbowtie", and paired = 'fr'. Methylation levels were determined using the qMeth function from the QuasR package.

**Amplicon and SureSelect bisulfite data processing**. Methylation levels for CpGs in amplicons was determined using the AmpliconViews function from the R package AmpliconBiSeq[108] with the parameters 'conv = 80' and exp.var = '90'. A minimum of 100× coverage was required for all methylation calls. If coverage did not reach this threshold for any time point, the respective methylation level was set to NA. We further removed all CpGs with NA at day 0 or with more than one NA during the time course across all replicates. In addition, we filtered out one amplicon (chr7:149767665-149767784) containing nine CpGs due to high variability in methylation calls. From a total of 588 CpGs initially quantified, this filtering procedure resulted in 405 CpGs. For SureSelect, a minimum of 50× coverage in all time points and replicates was used to filter CpGs for further analysis.

**A dynamical model for DNA methylation**. To enable inference of methylation and demethylation rates from the time-course data, we conceived a dynamical model for DNA methylation. This model is framed in the context of a simple chemical reaction with two rates: unmethylated cytosines are converted to methylated cytosines at a de novo methylation rate ($k_{me}$), while methylated cytosines are converted to unmethylated cytosines at a demethylation rate ($k_{de}$). This system can be described by two differential equations (DGLs) and further simplified into a single DGL (Fig. 1a). To simulate the loss of DNMT3 over time, we modified the DGL to gradually reduce the $k_{me}$ over the course of the experiment through an exponential dampening factor $\exp(-k_E)$. We set $k_E$ to a value of 0.5 considering various aspects of the experimental system. More specifically, we first assumed that the loss of methylation maintenance is considerably greater than de novo methylation. On a conceptual level, methylation levels will decrease 50% every cell cycle if maintenance is completely disrupted. Given a doubling time of 16 h, this would correspond to a rate of $\log(2)/(16/24) = 1.04$. Because both loss of RNA and protein is not instantaneous, we set the value of $k_E$ to 0.5.

As complete loss of maintenance is probably the most extreme case, the rates under investigation in this study are expected to be substantially <1.04. Nevertheless, to include more extreme cases we still considered rates of up to two and discretized $k_{de}$ and $k_{me}$ at steps of 10%, covering a dynamic range of three orders of magnitude across 80 steps. This resulted in a total of 6400 parameter combinations for $k_{de}$ and $k_{me}$. We next used a brute force approach to solve the DGL numerically for all possible parameter combinations using the R package deSolve[109]. Initial conditions were set such that the methylation levels at time = 0 were equivalent to the respective steady-state levels given the rates of methylation and demethylation ($M_{eq} = k_{me}/(k_{de} + k_{me})$).

Having generated the 6400 methylation traces, we added two types of errors to take into account further aspects of the system not covered by the DGL. For example, our genotyping data for DNMT3 loss showed that the genetic excision was not complete, retaining on average 8% of functional DNMT3 alleles. We simulated incomplete excision by mixing 8% of the methylation levels observed at day 0 with 92% of the simulated time course. Finally, we also considered bisulfite conversion and sequencing errors by assuming 99.75% efficiency and injected those effects into the simulated methylation traces using the formula $y = (1 - 0.0025 - 0.0025) \times x + 0.0025$. In summary, for all 6400 parameter combinations, this procedure produced 6400 methylation traces that we compared to real time-course data in order to infer rates of methylation and demethylation.

**Inference of amplicon methylation/demethylation rates**. To infer methylation/demethylation rates as well as confidence values for those rates, we coupled the dynamical DNA methylation model (which mimics the loss of DNMT3a/b over time) to a statistical error model. We chose to use a reparameterized beta-binomial error model that was successfully applied to bisulfite data in the past[110]. In contrast to the binomial distribution B($n,p$) that is governed by the two parameters n (number of trails) and $p$ (success probability), the beta-binomial distribution BB($n, p, \gamma$) has one additional parameter $\gamma$ to account for over-dispersion. Over-dispersion is critical in the case of the amplicon data due to an average of 4000× coverage per CpG. Given this depth of coverage, the theoretical error predicted by the binomial distribution would be much smaller than the actual error observed between replicate experiments. We determined biological variation by calculating the standard deviations of methylation levels observed in replicate experiments (at day 0) as a function of read coverage. Additionally, we stratified the data by mean methylation level because noise in bisulfite data is also a function of the mean. This showed that the beta-binomial distribution performed well in capturing the error observed in the amplicon bisulfite data (Supplementary Fig. 2c). To determine the optimal value for $\gamma$, we performed a parameter sweep, calculating the sum of squared errors to the actual data and selecting the minimum ($\gamma = 0.0055$; Supplementary Fig. 2b). Coupling this error model to the dynamical model for DNA methylation allowed us to use a statistical framework to obtain uncertainties for the inferred parameters. For a given combination of the two parameters $k_{de}$ and $k_{me}$, we calculated the probability of the data given the parameters $p$(data|parameters) using the respective simulated trace to set the parameter $p$ in the beta-binomial distribution and the read coverage to set the parameter $n$. To obtain a single probability for a given CpG across the time course we multiplied all probabilities obtained at 0, 4, 8, 10, 13, 17, and 29 days. Each biological replicate was fit separately in this manner. Having performed this calculation for all possible parameter combinations, we then used Bayes' theorem to calculate the probability of the parameters given the data $p$(parameters|data) assuming a uniform prior. We did so by renormalizing the probabilities obtained from the calculation of $p$(data|parameters) to a total of 1. As optimal parameters for $k_{de}$ and $k_{me}$, we used the maximum likelihood solution that we extracted from $p$(parameters|data). To determine credible intervals for $k_{de}$ and $k_{me}$, we first calculated the respective marginal probability density functions $p$($k_{de}$|data) and $p$($k_{me}$|data) and then determined the range that covered 95% of the area under the curve. Replicates were combined by calculating the median for all rates as well as the standard errors for the credible intervals (ci = sqrt($ci_1^2 + ci_2^2 + ci_3^2$)/3). We additionally used our inference procedure to identify CpGs for which rates could not be determined. This occurred when the probability density functions $p$($k_{de}$|data) or $p$($k_m$|data) showed substantial above zero densities at the borders of our parameter space, indicating that the optimal parameters lied outside of our parameter space. We thus

considered CpGs identifiable only if they showed <0.08 probability at any border for all the replicates.

**Comparison of wild-type and TTKO rates**. To determine if de novo methylation and passive demethylation rates are affected by TET activity, we conducted time-course experiments and amplicon bisulfite sequencing using the parental line with TET1/2/3 activity. This assay was otherwise identical to that performed in the TTKO line using amplicon bisulfite sequencing. While inspecting the methylation traces over the time course, we noticed a subtle difference between wild-type and TTKO lines in the mock treatment. While methylation measurements in the TTKO mock time course remained stable, there was a slight increase in methylation between days 4 and 8 measured in the wild-type mock treatment (Supplementary Fig. 4d). This effect was also present in Cre-treated samples, suggesting a secondary effect of the transduction protocol independent of Cre activity.

To systematically account for this, we normalized the Cre-treated time-course measurements to the corresponding mock-treated samples. This was done through creating a baseline for each CpG in the Cre-treated samples. These baselines were established by first binning CpGs at day 0 in increments of 10% (ten bins total) and calculating the average methylation level at each time point within the bin. The binning allows for more a more robust baseline estimation because we are averaging many CpGs around a similar steady state. We then divided each point in the baseline by the measurement at day 0 for the respective bin. These correction factors were used on the corresponding Cre-treated samples by multiplying every value in the time course by the respective baseline adjusted value. Applying this correction resulted in traces closely resembling the dynamics seen in the TTKO amplicon time course (compare Supplementary Fig. 4d and Fig. 1f). After normalizing the Cre-treated samples, we inferred the $k_{de}$ for the wild-type time course $k_{de}^{WT}$ in an identical manner to that described for the TTKO amplicon time course.

If TET activity does not effect de novo methylation and passive demethylation rates, we should be able to predict the demethylation rate in wild type using only steady-state measurements in both cell lines. Our model explicitly describes the relationship between demethylation and steady-state levels, and this relationship is mathematically derived in Supplementary Fig. 4c. Using this equation, we estimated the demethylation rate in wild type, namely $k_{de}^{WT^\wedge}$, for 405 CpGs using only steady-state methylation levels in the two genetic backgrounds. We then compared $k_{de}^{WT^\wedge}$ to rates inferred using the whole time course $k_{de}^{WT}$, as described above (Supplementary Fig. 4e). For the 155 CpGs, whereby we could estimate and infer demethylation rate robustly, our estimates accurately recapitulate methylation dynamics in the wild-type setting and these predictions are significantly more accurate than comparing $k_{de}^{WT}$ and $k_{de}^{TTKO}$ directly (Supplementary Fig. 2f).

To rule out the possibility that our estimations are a result of the mock normalization procedure described above, we inferred rates in the wild-type time course in the absence of normalization. Importantly, our rate predictions were unaffected by this normalization procedure, highlighting the robustness of our model and rate inference strategy (Supplementary Fig. 2e–h).

**Inference of SureSelect methylation/demethylation rates**. Methylation and demethylation rates were inferred the same way as for the amplicon bisulfite data using a binomial distribution for the error model instead of a beta-binomial. Due to the lower coverage of the SureSelect data, the beta-binomial was not necessary. A simple binomial error model performed well in capturing the variability observed between replicate experiments (Supplementary Fig. 3b).

**Calculation of the identifiable landscape**. To determine the identifiable parameter space, we performed parameter inference on the 6400 simulated methylation traces for which $k_{de}$ and $k_{me}$ are known. Using our inference procedure described above, we then asked to what degree all parameters can be recovered. In the beta-binomial error model, we set $n$ to the median coverage observed in the amplicon data ($n = 3997$). By calculating credible intervals for all parameter combinations, we were able to visualize the identifiable landscape. Because our experiments consisted of three biological replicates, we calculated standard errors of the credible intervals with $n = 3$ before plotting the heatmap. We identified parameter combinations that could not be determined as stated above considering a maximum of 0.05 probability at the border of the probability density functions for $p(k_{de}|data)$ and $p(k_{me}|data)$.

**ChIP-seq data processing**. The following published ChIP datasets were used in this study (GEO accessions): ADNP (GSM2582357), CTCF (ref. [5]; GSM747535); KLF4, SOX2, NANOG, and OCT4 (ref. [111]; GSM2417188, GSM2417144, GSM2417143, GSM2417187, and GSM2417142); NRF1 (ref. [103]; GSM1891642); p53 (ref. [112]; GSM647224); PRDM14 (ref. [113]; GSM623989), REST (ref. [114]; GSM671095); TCFCP2I1, ESRRB, and ZFX (ref. [64]; GSM288350, GSM288350, and GSM288352); and MAFK and ZC3H11A (ref. [115]; GSM1003809, GSM1003810).

Datasets were downloaded from GEO using the SRAdb R package[116] and aligned to the mm10 assembly of the mouse genome using Bowtie[117] within the QuasR[118] package. Bowtie was run using QuasR default parameters, returning only unique alignments. For each sample, the average fragment length was inferred directly from the data. This was done by determining the most frequent distance between the 5′ end of plus and minus strand reads on chromosome 1 with a distance interval spanning (read length +20) up to 500 bp. The lower limit of this interval was set significantly larger than the read length due to a second peak in the distance histogram at the exact read length in some samples, likely caused by a mapping artifact. The distance between pairs of reads with identical 5′ positions were counted only once to reduce potential amplification biases. All read counting in given genomic regions was done using the QuasR function *qCount*, whereby reads were shifted by half the estimated average fragment length determined above. For all replicates across TF datasets, peaks were identified using MACS2[119] under default parameters and with corresponding control samples as a background. Resulting peaks were then filtered requiring at least 80% mappability. Here, we define mappability as the fraction of all possible 25mers in a given region that are uniquely mappable using the alignment parameters above. Because the percentage of mappable bases in the genome changes in a minor way when increasing the read length under the given alignment parameters (74.9% for 25mers, 80% for 36mers, and 83.3% for 50mers, while 51 is the longest read length in the dataset), we do not believe that this choice of read length to define mappability has a significant effect on the presented results. The library-size normalized counts were determined as:

$$ns_{IP} = \min(N_{IP}, N_{control}) \times (n_{IP}/N_{IP}) \text{ and } ns_{control} = \min(N_{IP}, N_{control}) \times (n_{control}/N_{control})$$

Where $n_{IP}$ and $n_{control}$ are the raw counts per peak, and $N_{IP}$ and $N_{control}$ are the total number of reads mapping to the genome in the IP and control sample, respectively. Thus counts were in each case scaled down to the smaller library size. For each dataset, enrichment over input in peaks was defined as $\log2(ns_{IP} + 8) - \log2(ns_{control} + 8)$, using a pseudo-count of 8 to decrease noise levels in case of low read counts. Only peaks with a log2 enrichment of at least 1 were retained for further analysis. The 500 top-enriched peaks (or all peaks if there were fewer than 500 peaks) were used for de novo motif finding using HOMER[120]. HOMER was run using the function *findMotifsGenome.pl* using six different motif lengths (6, 10, 14, 18, and 22) and 200nt long sequences centered on each peak as input. For each dataset, the top-enriched motif was retained. The start or end positions of weight matrices were trimmed in cases where at least four consecutive positions had very low information content. The resulting weight matrices were compared to entries for the corresponding factors in either the HOMER database, the Jaspar database[121] the Encode factor book (for the datasets from Mouse ENCODE, www.factorbook.org) or the original publications to confirm similarity to the previously inferred weight matrices for each corresponding factor. In cases where replicate ChIP experiments produced matrices in the opposite orientation, matrices were reverse complemented so they all had the same orientation. Each inferred weight matrix was then used to scan the genome using the matchPWM function from Biostrings R package[122]. Matching sequences were determined by requiring a log2 (odds) score of at least 10 (in log2 scale) over a uniform background. In cases where two (or multiple) matches overlapped (ignoring their strands), only the match with the highest log(odds) score was retained. This is frequently the case for palindromic or nearly palindromic weight matrices, which often generate a match to both strands. Finally, for each dataset, log2 enrichments at the predicted sites were calculated by counting reads in a 201 bp window centered at the midpoint of each motif. In cases of multiple replicates for a given TF, we selected the replicate with the largest number of enriched motif-centered regions (which corresponds to the GEO accession above), after ensuring that all replicates showed similar patterns.

**Replication timing data**. Data for replication timing in murine ESCs was downloaded in processed form from the ENCODE consortium[115]. The specific accessions used in this work are ENCFF001JUP and ENCFF001JUQ.

**Reporting summary**. Further information on research design is available in the Nature Research Reporting Summary linked to this article.

## Data availability

Raw and processed sequencing data has been deposited in the Gene Expression Omnibus (GEO) database under the accession number GSE129470. All publicly available data sets used are referenced in the relevant methods section. All other relevant data supporting the key findings of this study are available within the article and its Supplementary Information files or from the corresponding author upon reasonable request. The source data underlying Fig. 3b and Supplementary Fig. 1e–h are provided as a Source data file. A reporting summary for this article is available as a Supplementary Information file.

## Code availability

Data analysis and graphical representation was performed using custom R scripts and publicly available packages as denoted in the text. All scripts are available upon request.

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

## Acknowledgements

We thankfully acknowledge En Li and Hiroyuki Sasaki for sharing the Dnmt3a and Dnmt3b floxed conditionally deficient mouse strains. We thank Matthew Lorincz, Michael Stadler, Mario Iurlaro, and members of the Schübeler group for feedback on project and manuscript. Research in the laboratory of D.S. and A.H.F.M.P. are supported by the Novartis Research Foundation, the European Research Council (ERC) under the European Union's Horizon research and innovation program (Grant agreement no. 667951—ReaDMe to D.S and ERC 695288—Totipotency to A.H.F.M.P.). D.S. acknowledges support by the Swiss National Sciences Foundation. A.K acknowledges support by a Swiss National Fund Ambizione grant (PZOOP3_161493). A.F. acknowledges support by the Marie-Curie Training Network "Nucleosome 4D".

## Author contributions

A.F. and D.S. initiated the study. P.A.G., A.F., and D.S. designed experiments. D.G. conceptualized the rate inference strategy and performed the rate modeling. P.A.G., D.G., and L.B. performed data analysis. P.A.G., A.F., L.H., and D.I. performed experiments. P.A.G., D.G., and S.S. determined data acquisition strategies. D.I. and A.K. established, and performed probe enrichment. A.H.F.M.P. and F.Z. derived ESC lines. F.E. provided transduction reagents. D.S. supervised this work. P.A.G., A.F., D.G., L.B., and D.S. wrote the manuscript with input from all authors.

## Competing Interests

The authors declare no competing interests.
