## [Peer Review File · Nature Communications]

REVIEWERS' COMMENTS:

Reviewer #1 (Remarks to the Author):

In my original review of the manuscript from Ginno et al, I was generally very positive about the quality of the data and the interpretations made. My biggest hesitation was rather how much novelty this study brings to the DNA methylation field. In their response, the authors argue:

“We hope he/she agrees that establishing an experimental and modeling approach to assign enzymatic rates at the individual CpG level in the context of native chromatin is in and of itself a significant achievement.”

This is undoubtedly true, and I believe this study will be of high value to the field. Moreover, the authors more than amply responded to my major concerns with regards to deficiencies in their experimental approach. One point that piqued my interest was that late-replicating sites (often enriched for H3K9 methylation) exhibit higher levels of maintenance methylation. This makes intuitive sense, given the binding activity of UHRF1. However, in the study I referenced from Tanay and colleagues, in human somatic cells late-replicating sites exhibit more DNA methylation erosion. I wonder if the authors of this study have any thoughts for this apparent discordance (fast-cycling cells versus less fast? High de novo activity vs less high?). Anyway, before I go on too long expounding about the results, I would like to offer my recommendation for publication in *Nature Communications*, which is a fine fit for the study, and will certainly allow for high visibility.

Minor point

The authors mention that they do not want to show their proteomic nor genomic data showing no effect on the expression of the DNA methylation maintenance pathway (Reviewer Response R5). I think it is at least worth mentioning the existence of this data in the text as data not shown, as it is an important control. Alternatively, they could show a western and RT-qPCR as they suggest.

Reviewer #2 (Remarks to the Author):

The authors have addressed all my points.

Reviewer #3 (Remarks to the Author):

R17. While the modeling referenced above and ours may seem to be overlapping to the reviewer, there are numerous differences both in approach and findings. The modeling of Rulands et al. is a theoretical assessment of how methylation activity could give rise to oscillations in methylation levels during the transition from naïve to primed ES cells. It is important that the biological variation modeled in this work is for the most part less than 3% of average methylation. In contrast, our work is focused on a very different aspect: assigning rates of activity for all 3 branches of the methylation machinery to individual CpGs in the mouse genome by means of genetic manipulation. This allows us to infer very specific properties of the individual components of the methylation machinery requiring a minimal set of assumptions. In contrast, the transition from 2i to serum is a much less controlled transition in terms of the molecular players at work.

The 3% the authors are referring to is a value obtained from the 2i release experiment in Rulands, which was from bulk sequencing.

However from Ruland's single-cell sequencing experiment in vivo (epiblast) they describe an amplitude of roughly 10-20%. In serum ESCs biological variability is larger in enhancers (80%) and again in the 20% region genome-wide. These are therefore substantial amounts of turnover, which moreover by genetic experiments were shown to depend on DNMT3s and TETs, so the findings in the Rulands study seem quite relevant to this current study and we feel should be discussed in the MS.

R18.

We would suggest a text clarification that includes part of the authors' explanation.

R19.

Would the authors then say they measure 'apparent' turnover? If that's the case it may be worth saying that at some point (perhaps introduction or early results part).

Ginno, Gaidatzis, Feldmann et al., point by point response

Reviewer #1 (Remarks to the Author):

In my original review of the manuscript from Ginno et al, I was generally very positive about the quality of the data and the interpretations made. My biggest hesitation was rather how much novelty this study brings to the DNA methylation field. In their response, the authors argue:

“We hope he/she agrees that establishing an experimental and modeling approach to assign enzymatic rates at the individual CpG level in the context of native chromatin is in and of itself a significant achievement.”

This is undoubtedly true, and I believe this study will be of high value to the field. Moreover, the authors more than amply responded to my major concerns with regards to deficiencies in their experimental approach. One point that piqued my interest was that late-replicating sites (often enriched for H3K9 methylation) exhibit higher levels of maintenance methylation. This makes intuitive sense, given the binding activity of UHRF1. However, in the study I referenced from Tanay and colleagues, in human somatic cells late-replicating sites exhibit more DNA methylation erosion. I wonder if the authors of this study have any thoughts for this apparent discordance (fast-cycling cells versus less fast? High de novo activity vs less high?). Anyway, before I go on too long expounding about the results, I would like to offer my recommendation for publication in Nature Communications, which is a fine fit for the study, and will certainly allow for high visibility.

Minor point

The authors mention that they do not want to show their proteomic nor genomic data showing no effect on the expression of the DNA methylation maintenance pathway (Reviewer Response R5). I think it is at least worth mentioning the existence of this data in the text as data not shown, as it is an important control. Alternatively, they could show a western and RT-qPCR as they suggest.

R1b. In our revised version we have included both western and RT-qPCR measurements of Dnmt1 and Uhrf1 in the TET triple knock-out line as well as TTKO cells where Dnmt3a/b have been deleted (penta-knockout). This data has been added to Supplemental Figs. 1g and h and addressed on page 4 of the text:

“Transcript and most importantly protein levels of DNMT1 and UHRF1 remained comparable upon loss of Tet1/2/3 and Dnmt3a/b, arguing that the maintenance machinery is intact in these genetic backgrounds (Supplemental Figure 1g,h).”

We have included the raw data for these panels in the Source Data file. We hope this satisfies the reviewer’s request.

Reviewer #2 (Remarks to the Author):

The authors have addressed all my points.

Reviewer #3 (Remarks to the Author):

R17. While the modeling referenced above and ours may seem to be overlapping to the reviewer, there are numerous differences both in approach and findings. The modeling of Rulands et al. is a theoretical assessment of how methylation activity could give rise to oscillations in methylation levels during the transition from naïve to primed ES cells. It is important that the biological variation modeled in this work is for the most part less than 3% of average methylation. In contrast, our work is focused on a very different aspect: assigning rates of activity for all 3 branches of the methylation machinery to individual CpGs in the mouse genome by means of genetic manipulation. This allows us to infer very specific properties of the individual components of the methylation machinery requiring a minimal set of assumptions. In contrast, the transition from 2i to serum is a much less controlled transition in terms of the molecular players at work.

The 3% the authors are referring to is a value obtained from the 2i release experiment in Rulands, which was from bulk sequencing.

However from Ruland's single-cell sequencing experiment in vivo (epiblast) they describe an amplitude of roughly 10-20%. In serum ESCs biological variability is larger in enhancers (80%) and again in the 20% region genome-wide. These are therefore substantial amounts of turnover, which moreover by genetic experiments were shown to depend on DNMT3s and TETs, so the findings in the Rulands study seem quite relevant to this current study and we feel should be discussed in the MS.

R2b. While the reviewer indeed has a point that both studies are modeling DNA methylation, we still find that differences in experimental approach and modeling, notwithstanding length restrictions of the manuscript preclude us from a sufficient contrast. We have cited the Rulands et al. work in the revised version and hope this is sufficient for the reviewer.

R18.

We would suggest a text clarification that includes part of the authors' explanation.

R3b. We agree that the use of further examples above can help to explain our modeling and how each element is considered. While length restrictions preclude us from numerous examples, we do describe 3 scenarios in detail in figure 1 to demonstrate how our modeling takes into account both steady states and rates. We have opted to allow publishing of the reviews for our work so readers can consult the example above. We hope this adequately addresses the reviewers request.

R19.

Would the authors then say they measure 'apparent' turnover? If that's the case it may be worth saying that at some point (perhaps introduction or early results part).

R4b. If so, we would then need to define "apparent", and as we do not assign a numerical value to turnover *per se* but use it in the conceptual sense, we find this goes beyond the scope of our article.